# Focal Cortical Dysplasia Type Ⅲ Related Medically Refractory Epilepsy: MRI Findings and Potential Predictors of Surgery Outcome

**DOI:** 10.3390/diagnostics11122225

**Published:** 2021-11-29

**Authors:** Xiaozhuan Wang, Dabiao Deng, Chengqian Zhou, Honglin Li, Xueqin Guan, Liguang Fang, Qinxin Cai, Wensheng Wang, Quan Zhou

**Affiliations:** 1Department of Radiology, Academy of Orthopedics Guangdong Province, The Third Affiliated Hospital of Southern Medical University, Guangzhou 510630, China; WXZ24568@163.com (X.W.); bluewind_ddb@tom.com (D.D.); lihonglin163@163.com (H.L.); yuki20161126@163.com (X.G.); fanglg09@163.com (L.F.); Teddymiao0211@163.com (Q.C.); 2Department of Radiology, Guangdong 999 Brain Hospital, Guangzhou 510510, China; 3Department of Psychiatry and Behavioral Science, School of Medicine, John Hopkins University, Baltimore, MD 21278, USA; czhou34@jhmi.edu

**Keywords:** FCD type III, magnetic resonance imaging, outcome, refractory epilepsy

## Abstract

This study aims to explore the relationship between neuropathologic and the post-surgical prognosis of focal cortical dysplasia (FCD) typed-Ⅲ-related medically refractory epilepsy. A total of 266 patients with FCD typed-Ⅲ-related medically refractory epilepsy were retrospectively studied. Presurgical clinical data, type of surgery, and postsurgical seizure outcome were analyzed. The minimum post-surgical follow-up was 1 year. A total of 266 patients of FCD type Ⅲ were included in this study and the median follow-up time was 30 months (range, 12~48 months). Age at onset ranged from 1.0 years to 58.0 years, with a median age of 12.5 years. The number of patients under 12 years old was 133 (50%) in patients with FCD type Ⅲ. A history of febrile seizures was present in 42 (15.8%) cases. In the entire postoperative period, 179 (67.3%) patients were seizure-free. Factors with *p* < 0.15 in univariate analysis, such as age of onset of epilepsy (*p* = 0.145), duration of epilepsy (*p* = 0.004), febrile seizures (*p* = 0.150), being MRI-negative (*p* = 0.056), seizure type (*p* = 0.145) and incomplete resection, were included in multivariate analysis. Multivariate analyses revealed that MRI-negative findings of FCD (OR 0.34, 95% CI 0.45–0.81, *p* = 0.015) and incomplete resection (OR 0.12, 95% CI 0.05–0.29, *p* < 0.001) are independent predictors of unfavorable seizure outcomes. MRI-negative finding of FCD lesions and incomplete resection were the most important predictive factors for poor seizure outcome in patients with FCD type Ⅲ.

## 1. Introduction

Efractory epilepsy is a medical problem worldwide and approximately one-half (46.5%) of all cases are caused by FCD [1,2,3,4,5]. In 2011, the International League Against Epilepsy (ILAE) described an international consensus of classification for FCD. The concept of FCD type Ⅲ can be defined as FCD type I adjacent to another principal lesion, such as hippocampal sclerosis (HS), low-grade developmental epilepsy-associated tumors (LDEATs), or vascular malformations [6]. The pathogenesis of FCD type Ⅲ is uncertain, because it is hard to tell whether the cortical dyslamination is primary or secondary [7,8].

Generally, MRI is the preferred imaging technique for detecting the structural basis of epilepsy [9,10]. In the T2W fluid-attenuated inversion recovery (T2W FLAIR) imaging sequence, the cortical and subcortical hyper-signal is easier to detect [11,12]. Neuroimaging abnormalities have been reported in patients with FCD, such as expand/atrophic cortical, indistinctness of the gray–white matter junction, hyperintensity changes, etc., but most of these reports are from patients with FCD type Ⅰ and/or type Ⅱ. Patients with identifiable lesions are more satisfactory than patients with MRI-negative findings on presurgical evaluation, despite the precise localization of focus on both scalp and invasive EEG monitoring [13,14,15].

Many factors, such as febrile convulsions (FS), seizure types, surgical procedures and incomplete resection have been repeatedly investigated to explore the relationship with surgical outcome, but the reported results are not univocal [16,17,18]. Existing evidence failed to highlight the clinical, neuroimaging, and electrophysiological profile discriminating patients with FCD type Ⅲ and the impact of co-occurring lesions on seizure outcome in patients with FCD type Ⅲ has yet to be fully elucidated [19]. The assessment of type Ⅲ contains low interobserver agreement. In this way, we summarize the data of patients with FCD type Ⅲ to explore the relationship among clinical features, preoperative MRI manifestations, pathology of FCD Ⅲ, and the prognosis of postoperative epilepsy.

## 2. Materials and Methods

### 2.1. Participants

The retrospective study was approved by the institutional review boards. Hence, the Institutional Review Board waived patient consent. We identified 575 patients with histologically confirmed FCD-related refractory epilepsy between December 2015 and October 2019. A total of 266 patients were confirmed to be FCD type-Ⅲ-related refractory epilepsy. Figure 1 shows a flowchart describing the research process. Baseline presurgical clinical information, MRI findings, surgical procedure, pathologic data, and postsurgical state were collected from the medical records. The clinical diagnose with FCD type Ⅲ was confirmed by pathologists and radiologists according to standard consensus by the International League Against Epilepsy (ILAE) Task Force [2].

### 2.2. MRI Acquisition

MRI was performed on a 1.5 T scanner (Philips Gyro Scan Inter 1.5T) with an eight-channel head coil. All patients underwent MRI examinations with standard brain MRI protocol including routine T1WI, T2WI, and T2W FLAIR sequence. Imaging protocols are detailed in Appendix A.

### 2.3. Patients Evaluation

Baseline clinical information, MRI findings, surgical procedure, pathologic data, and postsurgical state were collected from the medical records. The types of presurgical seizures are divided into focal onset, generalized onset, focal to bilateral tonic-clonic and unknown onset according to the International League Against Epilepsy (ILAE) Task Force on Classification and Terminology Guidelines [20]. The postsurgical states of seizures were recorded by the electronic medical record at 3 months, 6 months, and 1 year. The outcome at the last follow-up was assessed according to Engel’s classification, where patients were categorized as seizure-free (Class I/Ia) and seizures recurrence (II–VI). The standard of incomplete resection is that the resected area is discordant with the EEG results.

All original images were reviewed by two readers (D.D, with 8 years of experience in neuroimaging; H.L, with 6 years of experience in neuroimaging), who were blinded to the clinical, pathological information and surgical outcome, on each transverse section from T2W FLAIR sequence. When two physicians disagreed, a third radiologist with 30 years of experience in the field of neuroradiology gave the final opinion and reached a consensus. The MRI-negative is defined as invisible abnormal signals of cortical or subcortical. To assess the consistency and reproducibility between the two observers, the intraclass correlation coefficient (ICC) was calculated.

### 2.4. Statistical Analysis

The data storage and statistical analyses were performed with SPSS software (SPSS, version 20, IBM). Since these were all non-normally distributed variables, they were expressed as medians and IQ ranges. The Wilcoxon rank-sum test and chi-square analysis were used to analyze numerical variables and categorical variables separately following nonnormal distribution. In addition, odds ratios (ORs) and 95% confidence intervals (CIs) were calculated. To evaluate the effect of variables on postoperative seizure outcome, variables with a *p* ≦ 0.15 on univariate analysis were included in a multivariate Cox proportional hazards regression analysis.

## 3. Results

### 3.1. Clinical Data

A total of 266 patients of FCD type Ⅲ included in this study (171 men, 95 women; age range 6–19 years) were selected for analysis. The median follow-up time was 30 months (12–48 months). Age at onset ranged from 1.0 years to 58.0 years, with a median age of 12.5 years. The number of patients under 12 years old was 133 (50%). The duration of epilepsy symptoms ranged from 0.5 years to 51.0 years. A history of FS was present in 42 (15.8%) cases. In the entire postoperative period, 179 (67.3%) patients were seizure-free. Presurgical and surgical findings are described in Table 1.

### 3.2. Radiologic and Pathology Findings

Surgical pathology findings for 266 patients with FCD type III are shown in Table 2. According to the ILAE, 183 (68.8%), 61 (22.9%) and 22 (8.3%) cases were histopathologically diagnosed with FCD Ⅲa, FCD Ⅲb and FCD Ⅲc, respectively. Thickening/atrophic cortical were detected in 90 (13.5%) patients, the indistinctness of the gray–white matter junction in 176 (66.2%) patients, hyperintensity changes in 166 (62.4%) patients, and MRI negative findings of FCD in 36 (13.5%) patients. There were 156 (58.6%) cases with two or more combinations of cortical abnormalities. For FCD type Ⅲb, 41 (67.2%) were ganglioglioma (GG), 14 (22.9%) were dysembryoplastic neuroepithelial tumor (DNT), 1(1.6%) was papillary glioneuronal tumor (PGNT), 1(1.6%) was angiocentric gliomas (AG) and 1(1.6%) was GG, complicated by DNT. FCD type Ⅲc include 15 (68.2%) cases of cerebral cavernous hemangioma malformations (CCM) and 7 (31.8%) cases of cerebral arteriovenous malformations (AVM). FCD type Ⅲa was evaluated with 183 patients, of which 15 (8.6%) cases were FCD typeⅠa, 147 (84%) cases were FCD typeⅠb and 13 (7.4%) cases were FCD typeⅠc. We also found that 5 patients (8.6%) of FCD type Ⅲb were FCD typeⅠa, 49 (84.5%) were FCD typeⅠb, and 4 (6.9%) were FCDⅠc in 62 cases. In the FCD type Ⅲc as a whole, there was 1 case of FCD typeⅠa, and 21 cases of FCD typeⅠb, accounting for 4.5% and 95.5%, respectively. MRI findings of adjacent cortex in patients with FCD type Ⅲ are shown in Figure 2.

Typical imaging manifestations of FCD type Ⅲ are listed in Figure 3. The majority of tumors were located in the temporal lobe (*n* = 70, 84.3%) closing to the cortex, ten cases were located in the frontal lobe, two cases in the occipital lobe, and one case in the temporal occipital lobe. Four imaging patterns were differentiated in the low-grade, developmental, epilepsy-associated brain tumors, including a solid mass (*n* = 14, 22.9%), a predominantly cystic mass with an obvious mural nodule (*n* = 9, 14.8%), polycystic (*n* = 16, 26.2%), and mono-cystic (*n* = 22, 36%). Cerebral cavernous hemangioma malformations (*n* = 15, 68.2%) typically appear on T2W FLAIR sequences as a reticulated mixed-signal center with a surrounding hypointense rim, described as a popcorn pattern. A history of one case included an intracranial hemorrhage due to a cavernoma, which was surgically removed, but the magnetic resonance imaging of the brain revealed new cavernomas 1 year later, which were not identified in the previous imaging.

### 3.3. Predictive Variables for Postoperative Seizure-Freedom

In the univariate analysis, gender, seizure type, histology, FCD subtypes, lesion location, side of surgery, and procedure were not significant factors. The following factors were selected for multivariate analysis: age at epilepsy onset, duration of epilepsy, febrile convulsions, MRI-negative, complete removal, seizure type (*p* ≤ 0.15). After multivariate analysis, both being MRI-negative (*p* < 0.015) and incomplete resection (*p* < 0.001) decreased the likelihood that patients would be seizure-free after surgery. Table 3 shows the analysis results.

## 4. Discussion

FCD is a malformation caused by cortical development abnormalities characterized by cortical disorganization and the occurrence of dysmorphic cells, which can be the cause of drug-resistant epilepsy or intractable epilepsy. FCD type III is usually the onset of epilepsy in adolescence. The average age of onset of epilepsy in this study is 12.5 years old, and adolescents less than 12 years old account for 50% of the people in this study. Similar to previous reports, FCD type Ⅲa is the commonest pathology in FCD type Ⅲ [21]. Low-grade epilepsy-associated brain tumors (LEATs) are the second most common cause of drug-resistant epilepsy. In our study, GG and DNT accounted for 67.2% of all FCD type Ⅲ b, and they were the most common and representative tumor entities in both adults and children [22]. Epileptogenic foci mainly occur in the temporal lobe, but can also occur in the external temporal location. A total of 77.0% of FCD type Ⅲb is located in the temporal lobe, but the neurodevelopmental basis of its preference in the temporal lobe is unclear and requires further study. The pathological type of FCD type III in 217 cases was FCD Ib with other major lesions, which accounted for 85.1% of the people in this study.

The imaging findings of major lesions associated with FCD are easy to identify. The most common manifestations of hippocampal sclerosis are reduced hippocampal volume and/or increased signal intensity of the T2WI FLAIR sequence. T2WI FLAIR sequence can recognize the vast majority of hippocampal sclerosis, but there are still some patients with histopathologically confirmed hippocampal sclerosis. MRI cannot show clear morphological and signal abnormalities [23]. There are still some difficulties in the differential diagnosis of GG and DNT before operation. Both show clearly defined masses with cystic or solid components, some of which may be completely cystic or only local leukoaraiosis. Lesions are often located in the cortex or white matter, ranging in size from less than one millimeter to dozens of millimeters. Cerebral cavernous hemangioma and arteriovenous malformations are the most common pathological types in FCDIII c, of which cavernous hemangioma is more common, accounting for 63.6% of patients in this study. CCM has typical imaging findings: due to the rupture and bleeding of the thin blood vessel wall, hemosiderin is progressively deposited in the surrounding brain parenchyma, showing mixed high-signal intensity in the FLAIR sequence and “popcorn-like” changes, surrounded by low-signal hemosiderin bands [24]. One case of cavernous hemangioma recurred after surgery in our research; the behavior and appearance of this CCM should be considered naturally dynamic lesions. This also suggests that the order of occurrence of FCD and other combined major lesions needs further study [25].

Being MRI-negative is one of the risk factors for the poor prognosis of epilepsy. The main lesions associated with FCD can obviously find the corresponding signal abnormalities on imaging, but the morphology and signal of the surrounding cortex are often slightly changed, resulting in missed diagnosis of cortical lesions [26,27]. The image changes of FCD are more hidden, and there are four main types of manifestation: expand/atrophic cortical, the indistinctness of the gray–white matter junction, hyperintensity changes and being MRI-negative. The most common imaging findings in this study were high signal intensity in the cortex and/or white matter and unclear demarcation between gray and white matter. These kinds of image manifestations often do not appear alone, and commonly occur in the form of two or more combinations. In this study, the pathological changes in 70.6% of MRI-negative patients were of the FCD Ib type, suggesting that there is a close relationship between MRI-negative manifestations and histopathological structure. The cellular density of the cortex is little altered, with disorganization in FCD type I, with microscopic pathology presenting a normal MRI imaging. The other combined lesions might affect the cerebral cortex and lead to subtle damage due to epileptic seizures or antiepileptic medication with no structural alterations, which are often difficult to detect using conventional MRI [28,29]. For example, FCD type Ⅲc perhaps leads to maturational disorder in the adjacent cortex with micro-columnar histological architecture due to chronic low-grade ischemia. MRI-negative findings are more closely linked to the underlying nature of structural MRI pathology. A visible lesion on MRI serves as a marker for resection, which is more likely to be co-located with the epileptogenic zone, and thus be closer to complete resection. Therefore, MRI-negative patients are more likely to have a poor prognosis after epilepsy resection.

For patients with FCD type Ⅲ, surgical resection of epileptogenic foci can benefit most patients with drug-resistant epilepsy. In previous studies, a ‘complete’ or ‘uncomplete’ resection was defined differently. Interictal EEG is highly sensitive to the localization of epileptogenic foci. In our study, complete resection are defined base on whether the extent of surgical resection is consistent with the abnormality of EEG. A total of 67.3% of patients were seizure-free after surgery in our study, leaving nearly 30% of patients with recurrent seizures. Obviously, this result indicates that there is another epileptogenic zone outside the resected area [30,31]. We found decreases in subcortical white matter fiber bundles, a decrease in perfusion, the aggregation of large neurons, and decreases in myelination where epileptogenic foci are located. An epileptogenic zone is complex, involving “normal-appearing” perilesional cortex in most cases. Epileptic networks are organized with irritative zone, which can cover the actual seizure zone, beyond the borders of the visible lesion, or to more distant zones [32,33]. Cortical gliosis outside the epileptogenic focus and the nerve tissue cells at the microstructure level are replaced by water. There are abnormal synaptic connections between abnormal tissues and normal tissues, as well as within abnormal tissues, thus forming an abnormal and excitatory neural network, which increases susceptibility to seizures. The complete removal of epileptogenic foci can better block the formation of the epileptic neural networks, and further reduce, or even prevent, seizures. The visible lesions on MR define the scope of the epileptogenic focus to some extent, which may be consistent with the actual epileptogenic area. The incomplete removal of epileptogenic foci leaves more epileptogenic areas. These two factors jointly affect the postoperative prognosis of drug-resistant epilepsy, and the incomplete resection of epileptogenic foci is the most important predictive factor affecting the prognosis of epileptic surgery.

It is mandatory to conduct a detailed presurgical patient evaluation and presurgical planning. In recent years, there have been considerable advances in the automated detection of focal epileptogenic abnormalities based on structural MRI scans. These approaches use a combination of postprocessing and artificial intelligence techniques to automatically delineate structural abnormalities. At present, the mainstream MRI post-processing techniques used to assist in the detection of the FCD include surface-based classification (SBC) techniques [34], Voxel-based morphometry (VBM) [35] and voxel-based MRI morphometry analysis program (MAP) [36]. These MRI post-processing techniques can extract morphological and tissue signal strength features from T1WI, T2WI and FLAIR images, and can be further used to train neural network classifiers. This can improve the detection efficiency of epileptic foci caused by FCD. In the absence of structural abnormalities that are detectable on MRI, the noninvasive method of Magnetic Source Imaging (MSI) provided valuable information regarding. The location and extent of the epileptogenic area determined by MSI was essentially identical to that determined directly through extra-operative electrocorticography (ECoG) [37]. Beyond that, intraoperative shear-wave elastography (SWE), sono-elastography and contrast enhanced US are advanced ultrasound techniques, which were preliminarily applied to FCD surgery [38,39]. These ultrasound techniques have already been shown to help detect epileptic focus. Intraoperative MRI (iMRI) was applied to attain a balance between achieving the ultimate goal of complete seizure freedom while minimizing postoperative neurological deficits [40]. The incorporation of iMRI aids in the achievement of greater rates of gross-total resection and complete seizure freedom.

This study had some limitations: First, this is a retrospective study, and there may be a selection bias. Second, the sequence is relatively singlular. Although the sequence is sensitive to signal changes, it is still insufficient for detecting the thickness of the cortex and the morphological changes in the abnormal sulcus gyrus. Third, we failed to evaluate the neuropsychological outcomes, which are important aspects of surgical decision-making. With the extension of the observation time of the subjects, some patients with postoperative epilepsy without seizures may have seizures. Fifth, there is no stratification study on the FCDIII subtypes.

## 5. Conclusions

Overall, patients with FCD type Ⅲ have favorable outcomes after epilepsy surgery. T2-FLAIR provides diagnostic information prior to surgery, while clinical applications may provide information necessary for early patient consultation and postoperative management of epilepsy. We identified that MRI-negative findings of FCD lesions and incomplete resection were predictive variables of seizures after surgery.

## Figures and Tables

**Figure 1 diagnostics-11-02225-f001:**
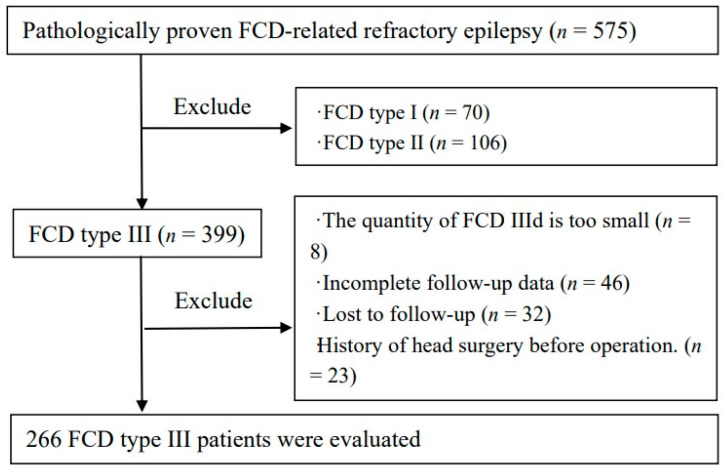
Inclusion and exclusion flowchart.

**Figure 2 diagnostics-11-02225-f002:**
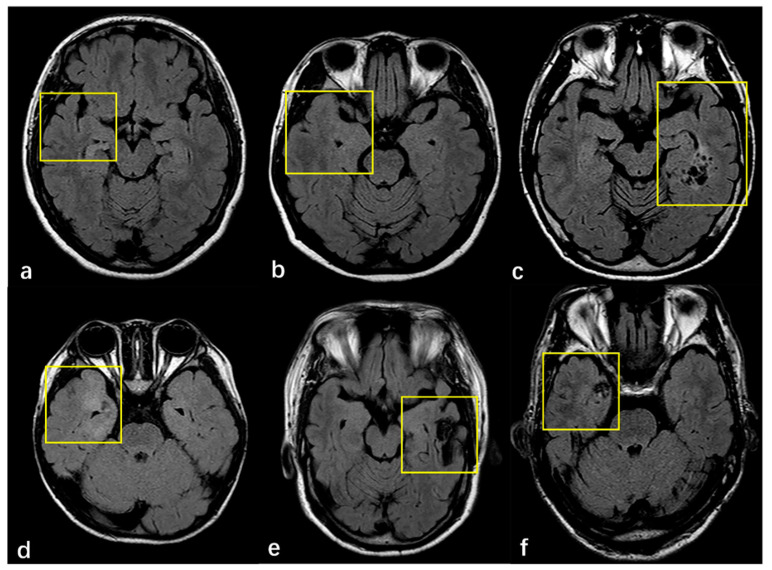
(**a**–**f**) MRI finding of adjacent cortex in patients with FCD. (**a**) Right hippocampal atrophy without visually adjacent cortex abnormality. (**b**) Right hippocampal hyperintense with the thickened cortex of ipsilateral temporal lobe. (**c**) GG is located medial to the left temporal lobe with a normal adjacent cortex. (**d**) The right temporal pole and medial temporal lobe cortex were thickened, and indistinctness of the gray–white matter junction was complicated by GG. (**e**) Right frontal AVM without visual alterations in the adjacent cortex. (**f**) There is a CCM in the right temporal lobe with adjacent cortex hyperintense and indistinctness of the gray–white matter junction.

**Figure 3 diagnostics-11-02225-f003:**
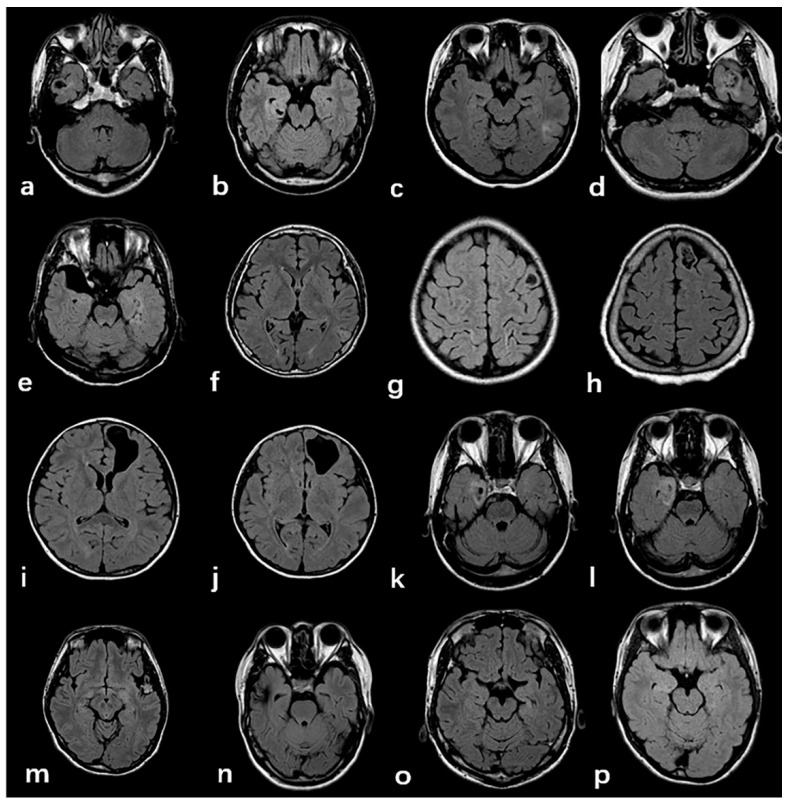
Typical imaging manifestations of FCD type Ⅲ. (**a**–**d**):GG; Figure (**e**–**h**): DNET; (**i**–**l**): PGNT; (**k**–**l**): AG. The above low-grade developmental epilepsy related tumors were mostly found in the temporal lobe, adjacent to the cortex, and dominated by cystic components. (**m**): CCM, typical “popcorn” change. (**n**): AVM, particulate hypointensity in the proximal cortex; (**o**,**p**): HS, characterized by hippocampal atrophy and increased signal intensity.

**Table 1 diagnostics-11-02225-t001:** Demographic Date for 266 Patients.

	Total	Class 1/1a	Class 2–6	*p*-Value
Age at epilepsy onset *	12.5 (6.0–19.0)	13 (7–20)	11 (4–18)	0.145
Duration of epilepsy *	10 (4–16)	9 (3–15)	12 (6–18)	0.004
Gender *n* (%)				0.340
Females	95 (35.7%)	52 (59.8%)	35 (40.2%)	
Males	171 (64.3%)	119 (66.5%)	62 (71.3%)	
Children (≤12 y)	133 (50.0%)	84 (46.9%)	49 (56.3%)	0.191
FS	42 (15.8%)	24 (13.5%)	18 (20.9%)	0.150
Seizure types				0.314
Focal onset	51 (19.2)	32 (17.9%)	19(21.8%)	
Generalized onset	139 (52.3%)	96 (53.6%)	43 (49.4%)	
Focal to bilateral tonic-clonic	63 (23.7%)	40 (22.3%)	23 (26.4%)	
Unknown onset	13 (4.9%)	11 (6.1%)	2 (2.3%)	
Histology				0.288
FCD Ⅲa	183 (68.8%)	118 (65.9%)	65 (74.7%)	
FCD Ⅲb	61 (22.9%)	46 (25.7%)	15 (17.2%)	
FCD Ⅲc	22 (8.3%)	15 (8.4%)	7 (8.0%)	
FCD				0.904
Ia	21 (8.2%)	15 (8.7%)	6 (7.3%)	0.932
Ib	217 (85.1%)	147 (85%)	70 (85.4%)	
Ic	17 (6.7%)	11 (6.4%)	6 (7.3%)	
Lesion location				
Temporal lobe	229 (86.1%)	15 (86.6%)	74 (85.1%)	
Extre-temporal lobe	12 (4.5%)	8 (4.5%)	4 (4.6%)	
Multiple lesion	25 (9.4%)	16 (8.9%)	9 (10.3%)	
Side of surgery				0.896
Left	118 (44.4%)	80 (44.7%)	38 (43.7%)	
Right	148 (55.6%)	99 (55.3%)	49 (56.3%)	
Procedures				0.858
Temporal lesionectomy	172 (64.7%)	11 (64.2%)	57 (65.5%)	
Anterior temporal lobectomy	51 (19.2%)	34 (19.0%)	17 (19.5%)	
Extratemporal lobectomy	10 (3.8%)	8 (4.5%)	2 (2.3%)	
Multilobar lesionectomy	33 (12.4%)	22 (12.3%)	11 (12.6%)	
Incomplete resection	40 (15.0%)	12 (6.7%)	28 (32.2%)	0.000
MRI features of FCD				0.680
Cortex atrophy/thickening	90 (13.5%)	59 (13.4%)	31 (13.8%)	
Indistinctness of the gray–white matter junction	176 (66.2%)	118(65.9%)	58 (66.7%)	1.000
Intensity	166 (62.4%)	116 (64.8%)	50 (57.5%)	0.281
MRI Negative	36 (13.5%)	19 (10.6%)	17 (19.5%)	0.056

Note: data are numbers of patients with percentages in parentheses. * Data are medians, with interquartile range in parentheses (IQR).

**Table 2 diagnostics-11-02225-t002:** Surgical pathology findings of 266 patients with FCD type III.

	Total	FCD Subtype			FCD-Associated Tumor						
		FCD Ia	FCD Ib	FCD Ic	GG	DNT	PNT	AG	GG+DNT	CCM	AVM
FCD IIIa	183	15 (8.6%)	147 (84.0%)	13 (7.4%)	/	/	/	/	/	/	/
FCD IIIb	61	5 (8.6%)	49 (84.5%)	4 (6.9%)	41 (67.2%)	14 (22.9%)	1 (1.6%)	1 (1.6%)	1 (1.6%)	/	/
FCD IIIc	22	1 (4.5%)	21 (95.5%)	0	/	/	/	/	/	15 (68.2%)	7 (31.8%)

**Table 3 diagnostics-11-02225-t003:** Logistic regression analysis of risk factors for FCD.

Factors	OR (95% CI)	*p* Value
Age at epilepsy onset	0.99 (0.96–1.03)	0.617
Duration of epilepsy	1.03 (0.99–1.07)	0.132
Febrile convulsions	0.73 (0.32–1.64)	0.441
MRI Negative	0.34 (0.45–0.81)	0.015
Complete removed	0.12 (0.05–0.29)	<0.001
Seizure Types		
Focal onset	1	0.394
Generalized onset	1.19 (0.20–6.90)	0.850
Focal to bilateral tonic-clonic	2.12 (0.42–10.9)	0.364
Unknown onset	2.47 (0.32–1.63)	0.295

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
