# Peer review of "Focal Cortical Dysplasia Type Ⅲ Related Medically Refractory Epilepsy: MRI Findings and Potential Predictors of Surgery Outcome"

_diagnostics, 2021, doi:10.3390/diagnostics11122225_

Round 1
Reviewer 1 Report
The authors report a large series of patients with drug-resistant epilepsy associated with FCD type III.
They summarized the data of patients to analyze the relationship among clinical features, preoperative MRI findings, pathology of FCD â…¢ and the postoperative outcome of epilepsy.
This article is interesting and well-written.
I have just few concerns to address before this paper is suitable for publication.
- the text contains some typos / grammatical errors.
- In the discussion section, as incomplete resection of the FCD was one of the most important predictive factor for poor seizure outcome, the authors should discuss about the different innovative tools which allow to improve the FCD intraoperative detection and resection, such as, elastrography (strain elastography: Prada et al, Clin Neurol Neurosurg, 2020; shear wave elastography: Mathon et al, J Clin Med, 2021), MRI (Sacino et al, Neurosurg Focus, 2016) and electrocorticography (Ishibashi et al, Neurol Res, 2002).
Reviewer 2 Report
The authors describe a very large series of patients with a combination of FCD and second pathologies (FCD III). Their cases are consistent with the current classification https://doi.org/10.1111/nan.12462 , but perhaps does not reflect some of the uncertainties in that classification, particularly in type IIIa. As all such series, it is retrospective, which they acknowledge, with the inherent uncertainties. The gender ratio of nearly twice as many males, is surprising (although some male predominance is seen in other studies) and merits some explanation. The surgical outcome of these cases is good when both lesion and FCD were resected and is similar to other forms of FCD. Incomplete resection of FCD was a negative predictor. My main concerns are stylistic. The English needs to be improved in a number of areas. Section 3.2 contains much useful information but is very difficult to work through and would be better presented as another table.
